# N-N(+) Bond-Forming Intramolecular Cyclization of O-Tosyloxy β-Aminopropioamidoximes and Ion Exchange Reaction for the Synthesis of 2-Aminospiropyrazolilammonium Chlorides and Hexafluorophosphates

**DOI:** 10.3390/ijms241411315

**Published:** 2023-07-11

**Authors:** Lyudmila Kayukova, Anna Vologzhanina, Pavel Dorovatovskii, Elmira Yergaliyeva, Asem Uzakova, Aidana Duisenali

**Affiliations:** 1JSC A. B. Bekturov Institute of Chemical Sciences, 106 Shokan Ualikhanov St., Almaty 050010, Kazakhstan; erg_el@mail.ru (E.Y.); a7_uzakova@mail.ru (A.U.); duisenali-a@mail.ru (A.D.); 2A. N. Nesmeyanov Institute of Organoelement Compounds, Russian Academy of Sciences, 28 Vavilova St., B-334, Moscow 119334, Russia; 3NRC “Kurchatov Institute”, 1 Kurchatova Pl, Moscow 123098, Russia; paulgemini@mail.ru

**Keywords:** chlorides, hexafluorophosphates, spiropyrazolines, toluenesulfonates, X-ray diffraction analysis, in vitro biological screening

## Abstract

Our research area is related to the spiropyrazolinium-containingcompounds, which are insufficiently studied compared with pyrazoline-containing compounds. Nitrogen-containing azoniaspiromolecules have also been well studied. In drug design and other areas, they are a priori important structures, since rigid spirocyclic scaffolds with the reduced conformational entropy are able to organize a closely spaced area. Azoniaspirostructures are currently of wide practical interest as ionic liquids, current sources (membranes), structure-directing agents in organocatalysis, and in the synthesis of ordered ceramics. Our goal was the synthesis of 2-aminospiropyrazolilammonium chlorides and hexafluorophosphates. Our methodology is based on the tosylation of β-aminopropioamidoximes with six-membered N-heterocycles (piperidine, morpholine, thiomorpholine, and phenylpiperazine) at the β-position. 2-Aminospiropyrazolilammonium chlorides and hexafluorophosphates were obtained by the reaction of double ion substitution in the reaction of toluenesulfonates of 2-aminospiropyrazolinium compounds with an ethereal solution of HCl in ethanol and with ammonium hexafluorophosphate in ethanol in quantitative yields of 55**–**97%. The physicochemical characteristics of the synthesized compounds and their IR and NMR spectra are presented. The obtained salts were additionally characterized by the single-crystal XRD analysis. The presence of both axial and equatorial conformations of spirocations in solids was confirmed. 2-Aminospiropyrazolilammonium chlorides and hexafluorophosphates have weak in vitro antimicrobial activity on Gram-positive and Gram-negative bacterial lines.

## 1. Introduction

Our research area concerns spiropyrazolinium-containing compounds, which are insufficiently studied as compared with other azoniaspiroalkanes (Figure 1). The latter possess numerous practical applications, which are described below. At the same time, pyrazoline derivatives, which belong to the family of nitrogen-containing heterocycles, have been the subject of a global diphenylpyrazoline market report by Market Strides (a global aggregator and publisher of market research reports). The diphenylpyrazoline market utilizes both pharmaceutical and industrial applications. The pharmaceutical industry is the largest end-user of diphenyl pyrazoline due to its various biological activities, including antioxidant, anti-inflammatory, anti-cancer, and anti-diabetic properties, as well as its ability to treat cardiovascular disorders. The industrial market covers textiles, detergents, paper production, cosmetics, and plastics. In addition, diphenyl pyrazoline is also used in the agrochemical industry as a pesticide and an insecticide [1].

It should be taken into account that the related family, azoniaspirocompounds, is a developed class of compounds with long-established synthetic methods and wide application areas. Thus, azoniaspiro compounds can be synthesized by reaction of the α,ω-dibromoalkane with a cyclic amine (e.g., 1,5-dibromopentane and piperidine to obtain 6-azonia-spiro-[5,5]-undecane [2]. Chiral quaternary N-spiroammonium bromides with 3′,4′-dihydro-1′H-spiro[isoindoline-2,2′-isoquinoline]skeleton were synthesized by reacting amino alcohols with the corresponding dibromide in acetonitrile or dichloromethane in the presence of DIPEA as a base at r.t. [3]. The spirocyclic ammonium derivatives contain a positively charged quaternary nitrogen atom, hence the fact that they can be isolated and characterized only in the form of salt [4]. Among spirocyclic nitrogen-containing compounds, small azoniaspiromolecules a priori represent important structures in drug design, since rigid spirocyclic scaffolds with reduced conformational entropy are able to selectively interact with active sites of pathogens and are expected to be selective drugs when bound to a target protein [5].

In this regard, the introduction mainly summarizes data on azoniaspirocompounds with parallel use of known data on spiropyrazolinium compounds, which may shed light on the possible areas of application of the 2-aminospiropyrazolilammoniums that we are developing.

### 1.1. Nitrogen-Containing Spiro-Heterocycles as Active Pharmaceutical Ingredients

Nitrogen-containing heterocyclic fragments provide biological activity to drug-like molecules. Most of the drugs on the pharmaceutical market include nitrogen-containing heterocyclic moieties [6]. The known nitrogen-containing spiro-heterocycles that are used in medical practice [7,8,9,10] include *fenspiride*, which is an anti-inflammatory, antiallergic, and antispasmodic agent, used for acute respiratory infections in most cases for children [7], and *irbesartan*, which is a selective angiotensin II receptor (type AT1) antagonist that reduces the concentration of aldosterone in the blood plasma, lowers peripheral vascular resistance, and lowers systemic arterial blood pressure (BP) [8]. *Fluspirilene* is an active antipsychotic agent with a pronounced antipsychotic effect, effective for hallucinations, delusions, and autism. It also calms emotional and psychomotor agitation [9]. *Trospium* is is a non-selective muscarinic receptor antagonist, which is used to treat the symptoms of an overactive bladder [10].

The results of the antibacterial activity study of chiral quaternary N-spiro ammonium bromides with 3′,4′-dihydro-1′H-spiro[isoindoline-2,2′-isoquinoline]skeleton showed that the Gram-negative bacteria were more susceptible towards the tested compounds than the Gram-positive bacteria. Their MICs on Gram-negative bacteria typically range from 12 to 100 mg/L, whilst those for Gram-positive bacteria vary from 50 to 200 mg/L [3].

As we have found, spiropyrazolilammonium compounds with the azoniaspiro moiety (Figure 2) exhibit high antitubercular and antidiabetic activity, which is nearly as large as those for the reference drugs rifampicin and acarbose [11,12].

### 1.2. Azoniaspirocompounds as a Renewable Energy Sources

The stability of anion-exchange membranes (AEMs) for renewable energy sources and the search for durable materials for them is an important issue, and research is being conducted in this direction. Azoniaspirocompounds have been a popular subject of research that is devoted to components of anion exchange membranes [14,15,16,17]. In particular, 6-azonia-spiro[5.5]undecane exhibits a much longer half-life (87.3 h and 110 h) compared to the half-lives of tetramethyl ammonium cation (61.9 h) and benzyltrimethyl ammonium cation (4.2 h) under the same testing conditions (6 M NaOH, 160 °C) [16]. When studying the effect of the ring size on membrane properties, it was found that both 6-azonia-spiro[5.5]decane and 5-azonia-spiro[4.4]nonan-based AEMs exhibit excellent alkaline stability (around 95% retention of conductivity after immersion in 1 M KOH solution at 80 °C for 720 h), high conductivity (up to 85.7 mS cmat 80 °C), and feasible tensile strength and elongation at break [17].

### 1.3. Azoniaspirocompounds as Phase Transfer Catalysts

In organic synthesis, the necessity of carrying out reactions between water soluble and oil soluble reagents arises very often. Starks found that organic-soluble quaternary ammonium cations were suitable agents for the transport of anions from the aqueous phase to the organic phase. They act as powerful reaction accelerators [18]. Kitamura et al. reported C2-symmetric ammonium salts (Maruoka Phase Transfer Catalysts)catalyzing monoalkylation of glycine-derived Schiff bases with alkyl halides in order to synthesize 𝛼-alkyl-𝛼-amino acids under remarkably low catalyst loadings [19,20,21,22].

### 1.4. Azoniaspirocompounds as Structure-Directing Agents in Zeolite Synthesis

Azoniaspirocompounds have found application as structure-directing agents (SDA) in the synthesis of zeolites, which have enormous use in oil refining and petrochemistry due to their exceptional catalytic and selective adsorption properties in combination with thermal and chemical stability. Zeolites also found numerous applications in fine chemical synthesis and environmental catalysis, and they dominate in the global catalyst market [23,24].

Every year, due to many synthetic methods to obtain different structures with variations in pore size, surface area, pore volume, and physical properties, the number of new zeolite structures increases, although the overall number of known topologies stored in a database created by the International Zeolite Association is only 252 [25].

The use of organic SDAs is associated with the concept of the host-guest chemistry of organic molecules and inorganic systems during zeolite synthesis [26]. The addition of these organic molecules to the reaction mixtures provokes a particular ordering of inorganic units around them, which directs the crystallization pathway towards a unique zeolite framework. Their molecular size and shape, hydrophobicity, rigidity vs. flexibility, and hydrothermal stability all determine the structure-directing effect of these organic species [27]. One of the latest requests in material chemistry concerns attempts to transfer the asymmetric nature of organic chiral molecules used as SDA to the zeolite lattice in order to produce chiral enantioselective frameworks [28].

Organic SDAs must meet the appropriate criteria for hydrophobicity/hydrophilicity due to the C/N+ ratio, typically between 11 and 16, and this limits the number of organic SDA candidates that can be used [29].

Hydrothermal synthesis of zeolites was carried out using the 5-azoniaspiro[4,4]nonane as SDA in a fluoride-free medium. Germanosilicate zeolites have received great attention due to a number of extra-large pore systems. Phase pure IWW-type zeolite was obtained after hydrothermal treatment at 175 °C for 3 days with a Si/Ge ratio around 4.9 [30].

A number of azoniaspiro compounds, namely, 5-azoniaspiro-[4,5]-decane, 6-azoniaspiro-[5,5]-undecane, and 6-azonia-spiro-[5,6]-dodecane, have all contributed to the selective formation of pure zeolites with given topologies [31]. The extra-large-pore germanosilicates have been synthesized in basic media using a wide variety of sterically overloaded spiroazonia compounds as SDAs. The influence of the composition of the reaction mixture and the nature of the SDAs (structure, hydrophilicity/hydrophobicity balance, rigidity, and pKa) on the phase selectivity and the degree of crystallinity has been investigated [32].

### 1.5. Azoniaspirocompounds as Ionic Liquids

The ability of room temperature ionic liquids to melt at the room temperature often suffers from poor thermal stability, and these room temperature ionic liquids have limited utility at higher temperatures [33]. Fullyinorganic molten salts are stable at elevated temperatures, but they do have high melting points, which limit their utility at lower temperatures. It has been proposed that azoniaspiro salts may partially bridge this gap, creating a continuum of ionic liquids from room to elevated temperatures. Azoniaspiro salts have not yet been widely explored in the context of ionic liquid chemistry, though [3].

Based on the need to expand the temperature range in which ionic liquids can operate, a comparative study of the thermal stability of a series of alkylammonium chloride salts, incorporating one saturated ring of five, six, seven atoms and azoniaspirocompounds with two saturated rings of five, six, seven or eight atoms, was undertaken. By employing temperature-ramped thermogravimetric analysis (TGA), relative thermal stabilities of the chloride species were compared. The authors hypothesized that the inclusion of a second cyclic structure would enhance steric interactions around the ammonium nitrogen center, providing improved thermal stability of the azoniaspiroammonium salts.In fact, two-ring spirocyclic tetraalkylammonium chloride salts are, in general, more thermally stable than the single-ring analogues, exhibiting higher T_onset_ and T_start_ values [33]. It is worth mentioning here that the hexafluorophosphate anion is a constituent part of a series of novel hexafluorophosphate salts that are based on N,N-dialkylimidazolium and substituted N-alkylpyridinium cations, which display the behavior of ionic liquids at temperatures above their melting point [34].

## 2. Results and Discussion

### 2.1. Synthesis and Spectra

In order to obtain new compounds that potentially possess useful properties as active pharmaceutical ingredients and/or for possible industrial uses, such as phase transfer catalysts, ionic liquids, structure directing agents (SDAs), and components for ion exchange membranes, a number of 2-aminopyrazolilammonium salts have been obtained—namely, chlorides and hexafluorophosphates (**9–16**, Figure 3).

The synthesis of 2-aminospiropyrazolilammonium chlorides **9**–**12** at the first stage included tosylation of four β-aminopropioamidoximes, where the β-amino group contains piperidin-1-yl (**1**), morpholin-1-yl (**2**), thiomorpholin-1-yl (**3**), or 4 -phenylpiperizin-1-yl (**4**) in CHCl_3_ at r.t. in the presence of diisopropylethylamine for 15–20 h with TLC monitoring. A series of toluenesulfochlorides (**5–8**) has been described by our team previously [35,36]. The tosylation products isolated as white precipitates were recrystallized from i-PrOH in 45–65% yields.

The second stage of the salts **9–12** synthesis consists in ethereal HCl action until pH = 2 is reached on tosylates **5–8** ethanol solutions. The reaction of exchange of tosylate anion for chloride anion proceeds instantaneously—after dropping an ether solution of HCl to the tosylates **5–8** ethanol solution, white flakes of chlorides **9**–**12** are immediately formed, and after recrystallization from *i*-PrOH, their yields were 55–97% (Figure 3, Table 1).

It is worth noting that compounds **11** and **12,** as shown by the X-ray diffraction data, crystallize as hydrates, and salt **12** contains the 2-aminospiropyrazolilammonium dication (Figure 4, Section 2.3):

Using a shortened synthetic route, the formation of hexafluorophosphates **13–16,** without isolating the intermediate 2-aminospiropyrazolilammonium tosylates **5–8,** was carried out. In a one-pot reaction, a mixture of *para*-toluolsulfochloride, one of the substrates of β-aminopropioamidoximes (**1–4**), the Bu_3_N base, and ammonium hexafluorophosphate in CHCl_3_ solution (Figure 5) were allowed to react. The reaction time to obtain products **13**–**16** in a 55–90% quantity after recrystallization from i-PrOH was 24 h.

To our knowledge, information about 1,5-diazaspiro-1-en-5-ium spiropyrazolinium salts is limited by our publications. However, when using the tosylation reaction, stable indazole derivatives previously obtained—1,1–disubstituted indazolium hexafluorophosphates, which have a nitrogen atom at the head of the bridge [37].

It is assumed that 2**–**aminospiropyrazolilammoniumtosylates, which are more thermodynamically favorable compared to O-sulfochlorination products, are first formed in the reaction mixture [38]. In this case, the intramolecular reaction of amination of intermediate products of O-tosylation of β-aminopropioamidoximes (Figure 5, A) with nucleophilic attack of the lone electron pair of the amine nitrogen atom on the imine nitrogen atom of the amidoxime group occurs, followed by the elimination of the good-leaving tosylate group, and this leads to tosylates **5**–**8**.

The oxime N-O bond break is accompanied by the N-N(+) bond formation. As a result of the exchange reaction of the tosylate group for the hexafluorophosphate anion in the reaction mixture 2-aminospiropyrazolilammonium hexafluorophosphates **13**–**16** are formed. It is noteworthy that, as can be seen from Table 1, the melting points of a number of chlorides **9**–**12** are significantly higher than those of a number of corresponding hexafluorophosphates **13**–**16**(Δt_m.p_. **9** and **13**: ~97 °C; Δt_m.p_. **10** and **14**: ~78 °C;Δt_m.p_. **11** and **15**: ~108 °C; Δt_m.p_. **12** and **16**: ~48 °C).

It must be noted that compounds **10** and **11** were obtained earlier. First, 2-amino-8-oxa-1,5-diazaspiro[4.5]dec-1-en-5-ammonium chloride monohydrate (**10**·H_2_O) was obtained by the acid hydrolysis of 5-substituted phenyl-3-[β-(morpholin-1-yl)ethyl]-1,2,4-oxadiazoles, and it has an X-ray diffraction description (the structure was registered at the Cambridge Crystallographic Database under the number CCDC 2049801 [11]). Herein, we present the crystal structure of pure anhydrous **10**, which has not yet been published. Second, 2-amino-8-thia-1,5-diazaspiro[4.5]dec-1-en-5-ammonium chloride (**11**) was precipitated for the first time by acid hydrolysis of 5-substituted phenyl-3-[β-(thiomorpholine-1-yl)ethyl]-1,2,4-oxadiazoles [39], and then through the reaction of double substitution of ions by the reaction of 2-amino-8-thia-1,5-diazaspiro[4.5]dec-1-en-5-ammonium toluenesulfonate with DIPEA hydrochloride [12]. Its structure was deposited to the Cambridge Crystallographic Databaseunder the number CCDC 2106798.

Thus, crystal structures **9**, **10**, and **12**–**16** are new. In the IR spectra of 2-aminospiropyrazolilammonium salts (**9**–**16**), there are characteristic bands of stretching vibrations. The stretching vibrations of the NH_2_ group occur in the region of 3342–3500 cm^−1^, with those of bonds C=N – at 1642–1659 cm^−1^, and the hexafluorophosphate group has a characteristic stretching vibration band νP–F in the region of 836–839 cm^−1^.

In the ^1^H-NMR spectra of compounds **9**–**16,** protons NH_2_ are in the field δ 7.21–7.50 ppm. The protons of the α- and β-methylene groups of compounds **9**–**16** give triplet signals in the regions, respectively, of δ 3.04–3.17 ppm and δ 3.70–3.95 ppm.

The diastereotopic nature of the geminal protons of the methylene groups that are located at the ammonium nitrogen atom manifests itself in the ^1^H-NMR spectra of compounds **9** and **12**–**16** as pairs of multiplet signals with an intensity of two protons at δ: 3.35m (ax), 3.44m (eq) (**9**); 3.52m (ax), 3.98m (eq) (**12**); 3.38m (ax), 3.46m (eq) (**13**); 3.41m (ax), 3.65m (eq) (**14**); 3.64m (ax), 3.87m (eq) (**15**); and 3.49m (ax), 3.98m (eq) (**16**). This can be interpreted as the effect of the slow rotation of β-heterocycles, which allows the equatorial and axial protons to be noticed.

The signals of Csp^3^ and Csp^2^ carbon atoms in the ^13^C-NMR spectra of compounds **9**–**16** are also characteristic. Thus, the signals of carbon atoms of the C=N bond of the pyrazolinium ring in compounds **9**–**16** were recorded in the range δ 168.2–170.1 ppm, and signals of aromatic carbon atoms Csp^2^ of compounds **12** and **16** were found in the range δ 115.3–156.7 ppm.

The carbon atoms of the α- and β-methylene groups of compounds **9**, **10**, **12**–**14**, and **16** give resonance in the ranges of δ 31.5–32.5 ppm and 44.5–62.4 ppm, respectively, and the α-methylene group signal of compound **15** is due to the electron-donating nature of sulfur in the heterocycle at 23.1 ppm. For this reason, there is also a higher field position of the signals of methylene groups carbon atoms located at the sulfur atom of the thiomorpholine ring in compound **15** at δ 31.3 ppm compared with the position of the ^13^C NMR signals of methylene groups at the oxygen atom in compound **10** (60.9 ppm) andcompound **14** (62.2 ppm) as well as at the nitrogen atom in the compounds **12** (44.5 ppm) and **16** (45.7 ppm).

Carbon atoms of methylene groups at the ammonium spirocyclic nitrogen atom of compounds **9**,**10**, and **11**–**16** are presented in the range δ 62.9–64.7 ppm, and carbon atoms of the (CH_2_)_3_ group of piperidine derivatives **9** and **13** have signals at δ 21.0 and 21.9 ppm and 20.5 and 21.5 ppm.

It should be pointed out that the ^13^C NMR spectra of compounds **13**–**16** have a characteristic feature—in the region of δ 38.4–39.5 ppm, a multiplet signal appears due to the presence of fluorine in the molecules.

### 2.2. The In Vitro Antibacterial Activity of 2-Amino-1,5-diazaspiro[4.5]dec-1-en-5-ammoniumchlorides (***9***–***12***) and Hexafluorophosphates (***13***–***16***)

The antimicrobial activity of the samples of 2-amino-1,5-diazaspiro[4.5]dec-1-en-5-ammonium chlorides (**9**–**12**) and hexafluorophosphates (**13**–**16**) was investigated against Gram-positive bacteria *Staphylococcus aureus* and *Bacillus subtilis* as well as Gram-negative *Escherichia coli, Pseudomonas aeruginosa* strain, and to the yeast fungus *Candida albicans* by diffusion into agar (wells).

The reference drugs were gentamicin for bacteria and nystatin for the yeast fungus *Candida albicans*. Table 2 contains data about the antimicrobial activity of the studied samples.

We discovered that samples **11**, **13**, and **15** exhibited weak antimicrobial activity against the Gram-positive *Staphylococcus aureus* cell line (11.2–13.7 mm), and that gentamicin is active on this strain at 24.5 mm. The samples of compounds **9**, **11**, **12**, **14**, and **15** also showed weak antimicrobial activity against the Gram-negative *Escherichia coli* strain (11.3–13.5 mm). For comparison, the drug gentamicin has high activity, and it inhibits bacteria on a disk with a diameter of 26.4 mm. At the same time, sample **11** also shows weak antibacterial activity against the Gram-negative test *Pseudomonas aeruginosa* strain (11.5 mm) (gentamicin—20.2 mm). None of the compounds showed antifungal activity against *Candida albicans*.

The remaining test samples do not show antimicrobial activity against the presented test strains of microorganisms.

### 2.3. X-ray Diffraction

Single-crystal XRD study of salts **9**, **10**, and **12**–**16** obtained from the reaction mixtures confirmed the formation of spiropyrazolinium salts in all of the cases. Salts (C_7_H_14_N_3_S)Cl·H_2_O (**11**) and (C_13_H_20_N_4_)Cl_2_·H_2_O (**12**) crystallized as hydrates. Asymmetric units of other salts (**9**, **13–16**) contained only one cation and one anion. Salt (C_13_H_20_N_4_)Cl_2_·H_2_O (**12**) is the first representative of spiropyrazolinium dications characterized by X-ray diffraction, and it contains a protonated nitrogen atom in position 8, which was clearly seen on the residual density map. Asymmetric units of all of the known salts are depicted in Figure 1.

In all of the solids, the five-membered ring is found in the envelope conformation with the C(1) atom situated 0.05(1)–0.39(1) Å above the mean plane formed by four other atoms. The six-membered heterocycle realized the chair conformation. However, overall spiropyrazolinium conformations are different. In particular, the N(2) atom of the five-membered pyrazolinium cycle in chloride **9** and hexafluorophosphates **13**–**16** realized more thermodynamically stable axial conformation in respect to the six-membered cycle, while in the other solids, it realizes the equatorial conformation. The phenyl ring in **12** is nearly coplanar with the N(1)-N(2)-C(3)=C(4) plane; the angle between them is 1.82(6)° only. For the two symmetrically independent cations in **16,**the corresponding angle values are equal to 47.42(8) and 74.39(8)°.

Crystal structures of chlorides and hexafluorophosphates indicate the low likelihood of H-bond formation with watermolecules, as proposed by Delori et al. [40]. The likelihood of hydrogen-bond formation was estimated using the Mercury package [41]), as described by Galek et al. [42]) and Vologzhanina [43]). In Table 3, data for a pure 2-aminospiropyrazolinium chlorides and hexafluorophosphates are compared for anhydrous salts and their hydrates. The N–H…Hal bond is the most expected one for all cases. Its propensity is much higher than that for the N–H…O(water) bond, which is in accordance with rare occurrence of hydrates for this family of salts. However, the high propensity of O–H…Hal bonds demonstrates that water molecules can act as linkers between anions.

Experimentally observed H-bonded associates are in accord with results of these calculations. All anions take part in H-bonding in order to form the associates depicted in Figure 2. H-bonded tetramers were found in salts **9**, **15**, and **16**. Infinite chains were observed in solid **10**, **12**, **14**, and **16**. A total of **13** infinite layers were found, which realize the square-lattice topology of their underlying net. We should note that **16** contains two types of associates. For all cases, the amino group takes part in two hydrogen bonds, and chloride anions act as acceptors of one (**9**), two (**10**), or three (**12**) H-bonds. In addition to H-bonds, halogen atoms take part in weak C-H…Hal interactions and halogen bonds.

## 3. Materials and Methods

### 3.1. Synthesis

The reagents were purchased from different chemical suppliers and were purified before use. FTIR spectra were obtained on a Thermo Scientific Nicolet 5700 FTIR instrument (Thermo Fisher Scientific, Inc., Waltham, MA, USA) in KBr pellets. The ^1^H- and ^13^C NMR spectra were recorded in DMSO-d_6._The ^1^H- and ^13^CNMR spectra of compounds **9**–**16** were acquired on a Bruker Avance III 500 MHz NMR spectrometer (Bruker, BioSpin GMBH, Rheinstetten, Germany). The signals of the residual undeuterated solvents were used as a reference for the ^1^H-NMR (2.50 ppm) and ^13^C-NMR (39.5 ppm) spectra.

Elemental analysis was carried out on a CE440 elemental analyzer (Exeter Analytical, Inc., Shanghai, China). The melting points were determined in glass capillaries on a PTP(M) apparatus (Khimlabpribor, Klin, Russia). The reaction progress and purity of the obtained products were controlled using Sorbfifil (Sorbpolymer, Krasnodar, Russia) TLC plates coated with CTX-1A silica gel, with grain size 5–17 µm, containing UV-254 indicator. The eluent for TLC analysis was a mixture of benzene–EtOH, 1:3. The solvents for the synthesis, recrystallization, and TLC analysis (ethanol, 2-PrOH, benzene, DMF, and acetone) were purified according to the standard techniques.

#### 3.1.1. General Procedure for the Preparation of 2-Amino-1,5-diazaspiro[4.5]dec-1-en-5-ammonium Chlorides (**9**, **10**, **12**)

Ethereal HCl was added dropwise until pH = 2 was reached to a solution of 0.0015 mol of tosylates 5, 8 in 10 mL of anhydrous ethanol. Then, 20 mL of absolute ether was poured. The resulting white precipitates of chlorides **9**, **10**, **12** were obtained after recrystallization from i-PrOH in 68 and 70% yields.

*2-Amino-1,5-diazaspiro[4.5]dec-1-en-5-ium chloride (***9***).* To a solution of 0.49 g (0.0015 mol) of 2-amino-1,5-diazaspiro[4.5]dec-1-en-5-ium 4-methylbenzenesulfonate (**5**) in 10 mL of ethanol ethereal HCl was added until pH = 2 was reached, and then 20 mL of absolute ether was poured. After recrystallization of the resulting white precipitate from i-PrOH, we obtained 0.19 g (68%) of chloride **9**, m.p. 288–290 °C, *R*_f_ 0.01. IR (KBr, ν, cm^–1^): 1663 (C=N); 2848 (Csp^3^−H); 3341, 3400 (N–H)_2_. ^1^H-NMR (500 MHz, DMSO-d_6_): 1.59 m, 1.80 m, 1.81 m, [6H, (CH_2_)_3_], 3.12 (t, J = 7.0 Hz, 2H, α-CH_2_), 3.80 (t, J = 7.0 Hz, 2H, β-CH_2_), 3.35 [m, 2H, N(+)(CHax)_2_] and 3.44 [m, 2H, N(+)(CHeq)_2_], 7.24 (s, 2H, NH_2_). ^13^C-NMR (126 MHz, DMSO-d_6_): 21.0, 21.9, 31.5, 60.7, 64.3, 168.5. Anal. Calcd for C_8_H_16_ClN_3_, %: C, 50.66; H, 8.50. Found, %: C, 50.27; H, 8.96.

*Amino-8-oxa-1,5-diazaspiro[4.5]dec-1-en-5-ium chloride (***10***)*. To a solution of 0.49 g (0.0015 mol) of 2-amino-8-oxa-1,5-diazaspiro[4.5]dec-1-en-5-ium 4-methylbenzenesulfonate (**6**) in 10 mL of ethanol ethereal HCl was added until pH = 2 was reached, and then 20 mL of absolute ether was poured. After recrystallization of the resulting white precipitate from i-PrOH, we obtained 0.19 g (68%) of chloride **10**, m.p. 271 °C, *R*_f_ 0.01. IR (KBr, ν, cm^–1^): 1648 (C=N); 2927, 2957 (Csp^3^−H); 3438 (N–H)_2_. ^1^H-NMR (500 MHz, DMSO-d_6_): 3.04 (t, J = 7.0 Hz, 2H, α-CH2), 3.60 [m, 4H, O(CH_2_)_2_], 3.92 [m, 6H, β-CH_2_, N(+)(CH_2_)_2_], 7.47 (s, 2H, NH_2_). ^13^C-NMR (126 MHz, DMSO-d_6_):32.0, 60.9, 64.5, 170. Anal. Calcd for C_7_H_14_ClN_3_O (191.66), %: C, 48.34; H, 8.11. Found, %: C, 48.20; H, 7.75.

*Mono(2-amino-8-phenyl-1,5,8-triazaspiro[4.5]dec-1-en-5-ium) dichloride hydrate (***12***).* To a solution of 0.6 g (0.0015 mol) 2-amino-8-phenyl-1,5,8-triazaspiro[4.5]dec-1-en-5-ium 4-methylbenzenesulfonate (**8**) in 10 mL of ethanol ethereal HCl was added until pH = 2 was reached, and then 20 mL of absolute ether was poured in the reaction mixture. After recrystallization of the resulting white precipitate from i-PrOH, we obtained 0.30 g (70%) of chloride 12; m.p. 277–280 °C, *R*_f_ 0.01. IR (KBr, ν, cm–1): 1651 (C=N), 1600 (C=C), 2839, 2981, 3002 (Csp^3^−H), 3303 (Csp^2^−H), 3308, 3400 [N(-H)_2_]. ^1^H-NMR (500 MHz, DMSO-d_6_): 3.17 (t, J = 7.0 Hz, 2H, α-CH_2_), 3.95 (t, J = 7.0 Hz, 2H, β-CH_2_), 3.58 [m, 4H, N(CH_2_)_2_], 3.52 [m, 2H, N(+)(CHax)_2_] and 3.98 [m, 2H, N(+)(CHeq)_2_], 7.41 (s, 2H, NH2), 7.81−8.53 (m, 5H, C(sp^2^)H). ^13^C-NMR (126 MHz, DMSO-d_6_): 31.5, 44.5, 61.5, 62.9, 115.3, 120.4, 123.8, 156.7, 168.6. Anal. Calcd for C_13_H_22_N_4_Cl_2_O (320,24): C, 48.76; H, 6.61. Found: C, 48.63; H, 6.36.

#### 3.1.2. General Procedure for the Preparation of 2-Amino-1,5-diazaspiro[4.5]dec-1-ene-5-ammonium Hexafluorophosphates (**13**–**16**)

To a solution of 0.0020 mol of β-aminopropioamidoximes **1**–**4** in 20 mL of CH_2_Cl_2_ 0.0020 mol of Bu_3_N in 5 mL of CH_2_Cl_2_ and 0.0020 mol of ammonium hexafluorophosphate were added, one after the other. At r.t., with stirring, a solution of 0.0020 mol of *para*-toluenesulfonyl chloride in 10 mL of CH_2_Cl_2_ was dripped. The reaction mixture was stirred for 24 h at r.t. The course of the reaction was monitored by TLC. The formed white precipitates of hexafluorophosphates **13**–**16** after recrystallization from i-PrOH were isolated in a 55–90% yield.

*2-Amino-1,5-diazaspiro[4.5]dec-1-en-5-ium hexafluorophosphate*(**13**). To a solution of 0.34 g (0.002 mol) of β-(piperidin-1-yl)propioamidoxime(**1**) in 20 mL of CH_2_Cl_2_ 0.37 g (0.002 mol) of Bu_3_N in 5 mL of CH_2_Cl_2_ and 0.33 g (0.002 mol) of NH_4_F_6_ all of these elements were added, one after the other. Following stirring for 0.5 h, 0.38 g (0.002 mol) of *para*-touenesulfochloride in 10 mL of CH_2_Cl_2_ was dripped. The reaction mixture was stirred for 24 h at r.t. The progress of the reaction was monitored by TLC. After recrystallization of the resulting white precipitate from *i*-PrOH, 0.33 g (55%) of transparent crystals of hexafluorophosphate **13** were obtained, m.p.192–193 °C, *R*_f_ 0.05. IR (KBr, ν, cm^–1^): 837 (P-F), 1651 (C=N), 2949(Csp^3^−H), 3399, 3503 [N(-H)_2_]. ^1^H-NMR(500 MHz, DMSO-d_6_): 1.61 m, 1.75 m, 1.89 m, [6H, (CH_2_)_3_], 3.14(t, J = 7.0 Hz, 2H, α-CH_2_), 3.38 [m, 2H, N(+)(CHax)_2_] and 3.46 [m, 2H, N(+)(CHeq)_2_], 3.76(t, J = 7.0 Hz, 2H, β-CH_2_), 7.21(s, 2H, NH_2_). ^13^C-NMR (126 MHz, DMSO-d_6_): 20.5, 21.5, 31.11, 38.9 (m, F), 60.3, 64.1, 168.2.Anal. Calcd for C_8_H_16_F_6_N_3_P (299,20) C, 32.11; H, 5.39.Found: C, 32.63; H, 5.45.

*2-Amino-8-oxa-1,5-diazaspiro[4.5]dec-1-en-5-ium hexafluorophosphate (***14***).* To a solution of 0.35 g (0.002 mol) of β-(morpholin-1-yl)propioamidoxime(**2**) in 20 mL of CH_2_Cl**_2_** 0.37 g (0.002 mol) of Bu_3_N in 5 mL of CH_2_Cl_2_ and 0.33 g (0.002 mol) of NH_4_F_6_ all of these elements were added, one after the other. Following stirring for 0.5 h, 0.38 g (0.002 mol) of *para*-touenesulfochloride in 10 mL of CH_2_Cl_2_ was added. The reaction mixture was stirred for 24 h at r.t.The progress of the reaction was monitored by TLC. After recrystallization of the resulting white precipitate from *i*-PrOH, 0.54 g (90%) of transparent crystals of hexafluorophosphate **14** were obtained, m.p.191–193 °C, *R*_f_ 0.08. IR (KBr, ν, cm^–1^): 839 (P-F)_,_ 1119 (C-O), 1645 (C=N),2957(Csp^3^−H), 3399, 3505 [N(-H)_2_]. ^1^H-NMR(500 MHz, DMSO-d_6_): 3.14(t, J = 7.0 Hz, 2H, α-CH_2_), 3.76(t, J = 7.0 Hz, 2H, β-CH_2_), 3.38 [m, 4H, O(CH_2_)_2_], 3.41 [m, 2H, N(+)(CH_ax_)_2_] and 3.65[m, 2H, N(+)(CH_eq_)_2_], 7.30 (s, 2H, NH_2_). ^13^C-NMR (126 MHz, DMSO-d_6_): 31.3, 38.4 (m, F), 62.2, 63.2, 169.0.Anal. Calcd for C_7_H_14_F_6_N_3_OP (301,17) C, 27.92; H, 4.69.Found: C, 28.15; H, 4.96.

*Amino-8-thia-1,5-diazaspiro[4.5]dec-1-en-5-ium hexafluorophosphate (***15***).* To a solution of 0.38 g (0.002 mol) of β-(thiomorpholin-1-yl)propioamidoxime(**3**) in 20 mL of CH_2_Cl_2_0.37 g (0.002 mol) of Bu_3_N in 5 mL of CH_2_Cl_2_ and 0.33 g (0.002 mol) of NH_4_F_6_ all of these elements were added, one after the other. Following stirring for 0.5 h, 0.38 g (0.002 mol) of *para*-touenesulfochloride in 10 mL of CH_2_Cl_2_ was added. The reaction mixture was then stirred for 24 h at r.t. The progress of the reaction was monitored by TLC. After recrystallization of the resulting white precipitate from *i*-PrOH, 0.64 g (85%) of transparent crystals of hexafluorophosphate **15** were obtained, m.p.180–182 °C, *R*_f_ 0.01. IR (KBr, ν, cm^–1^): 841 (P-F)_,_1221 (C-S), 1647 (C=N), 3189 (Csp^3^−H), 3397, 3503 [N(-H)_2_]. ^1^H-NMR(500 MHz, DMSO-d_6_): 3.10 (t, J = 7.0 Hz, 2H, α-CH_2_), 3.70(t, J = 7.0 Hz, 2H, β-CH_2_), 2.97 [m, 2H, S(CHeq)_2_] and 3.11 [m, 2H, SCH(ax)2], 3.64 [m, 2H, N(+)(CHeq)_2_] and 3.87 [m, 2H, N(+)(CHax)_2_], 7.40 (s, 2H, NH_2_). ^13^C-NMR (126 MHz, DMSO-d_6_): 23.1, 31.3, 39.1 (m, F), 62.4, 64.7, 168.9.Anal. Calcd for C_7_H_14_F_6_N_3_PS (317,23) C, 26.50; H, 4.45.Found: C, 26.91; H, 4.25.

*2-Amino-8-phenyl-1,5,8-triazaspiro[4.5]dec-1-en-5-ium hexafluorophosphate (***16***).* To a solution of 0.50 g (0.002 mol) of β-(4-phenylpiperazin-1-yl)propioamidoxime(**4**) in 20 mL of CH_2_Cl_2_ 0.37 g (0.002 mol) of Bu_3_N in 5 mL of CH_2_Cl_2_ and 0.33 g (0.002 mol) of NH_4_F_6_ for 0.5 h, 0.38 g (0.002 mol) of *para*-toluenesulfochloride in 10 mL of CH_2_Cl_2_ was added. The reaction mixture was then stirred for 24 h at r.t. The progress of the reaction was monitored by TLC. After recrystallization of the resulting white precipitate from *i*-PrOH, 0.64 g (85%) of transparent crystals of hexafluorophosphate (**16**) were obtained, m.p. 240–242 °C, *R*_f_ 0.01. IR (KBr, ν, cm^–1^): 836 (P-F)_,_ 1250 (C-N), (1597 (C=C), 1642 (C=N),2863 (C_sp3_−H), 3313 (C_sp2_−H), 3411, 3515 [N(-H)_2_]. ^1^H-NMR (500 MHz, DMSO-*d*_6_): 3.17 (t, J = 7.0 Hz,2H, α-CH_2_), 3.95(t, J = 7.0 Hz, 2H, β-CH_2_), 3.56 [m, 4H, N(CH_2_)_2_], 3.49 [m, 2H, N(+)(CH_ax_)_2_] and 3.98[m, 2H, N(+)(CH_eq_)_2_], 7.29 (s, 2H, NH_2_), 6.91−7.13 (m, 5H, C_(sp2)_H). ^13^C-NMR (126 MHz, DMSO-*d*_6_): 32.5, 39.5 (m, F), 45.7, 62.6, 64.10, 117.5, 121.7, 130.7, 151.0, 170.1. Anal. Calcd for C_13_H_19_F_6_N_4_P (376,29): C, 41.50; H, 5.09. Found, %: C, 41.92; H, 5.51.

### 3.2. In Vitro Evaluation of Antimicrobial and Antifungal Activity of the Compounds ***9***–***16***

In vitro antimicrobial activity of eight samples **9**–**16** against *Staphylococcus aureus*, *Bacillus subtilis* strains of Gram-positive bacteria, *Escherichia coli* and *Pseudomonas aeruginosa* of Gram-negative strains, and *Candida albicans* yeast fungus was carried outby the agar diffusion method (wells). The reference drugs were gentamicin for bacteria and nystatin for the yeast fungus *Candida albicans* [44].

The cultures were grown in a liquid medium at pH 7.3 ± 0.2 at a temperature of 30 to 35 °C for 18–20 h. The cultures were diluted 1:1000 in a sterile 0.9% isotonic sodium chloride solution, 1 mLin cups with appropriate elective, nutrient media for the studied test strains, and inoculated according to the «solid lawn» method. After drying, wells 6.0 mm in size were formed on the agar surface, into which solutions of the studied samples, gentamicin, and nystatin were all added. In the control, distilled water was used in equivolume amounts.

The studied samples were tested in the amount of 1 μg. The inoculations were incubated at 37°C, and the growing cultures were counted after 24 h. The antimicrobial activity of the samples was assessed by the diameter of the growth inhibition zones of the test strains (mm). The diameter of growth inhibition zones less than 10 mm and continuous growth in the cup was assessed as the absence of antibacterial activity, 10–15 mm being weak activity, 15–20 mm being moderately pronounced activity, and more than 20 mm being pronounced activity. Each sample was tested in three parallel experiments. Statistical processing was carried out by parametric statistics methods with the calculation of the arithmetic mean and standard error.

### 3.3. X-ray Diffraction

Single crystals precipitated from reaction mixtures. Intensity data for **15** were measured at 100.0(2) K using a 1-axis MarDTB goniometerequipped with Rayonix SX165 CCD detector at the “Belok/XSA” beamline of the Kurchatov Synchrotron Radiation Source [45,46]. The direct geometry and φ-scanning with the detector plane perpendicular to the beam (λ = 0.745 Å) were used. The data were indexed and integrated using the XDS, ver. 2023 software [47]. The intensities of reflections for the other samples were measured with a Bruker Apex II DUO CCD diffractometer (λ(MoKα) = 0.71073Å, graphite monochromator) at 100.0(2) K (**12**·H_2_O), 140.0(2) K (**9**, **10**, **13**, **14**), or 295.0(2) K (**16**). The structures were solved using the SHELXT program [48] and refined against F^2^ using SHELXL-2018 [49] and OLEX2 [50] program packages. Non-hydrogen atoms were refined in an anisotropic approximation. Positions of H(C) atoms were calculated and H(O) atoms were localized on difference Fourier maps. Hydrogen atoms were refined isotropically.The riding model was applied with U_iso_(H) = 1.5U_eq_(O), 1.2U_eq_(N) and 1.2U_eq_(C). Experimental details and crystal parameters are listed in Appendix A. Crystallographic information files are available from the Cambridge Crystallographic Data Center upon request (http://www.ccdc.cam.ac.uk/structures, accessed on 20 May 2023).

## 4. Conclusions

Synthetic paths to 2-aminospiropyrazolilammonium chlorides and hexafluorophosphates have been attested. Target compounds were obtained with good yield and characterized by physical-chemical and spectral methods. XRD data demonstrated that the compounds are able to form different conformations, and that they readily take part in H-bonding through the amino-group. Hexafluorophosphates are characterized by their relatively low melting temperatures, which makes them potentially active as ionic liquid salts. The in vitro screening of the salts also revealed weak antimicrobial activity against the Gram-positive *Staphylococcus aureus* strain for **11**, **13**, and **15**, and against the Gram-negative test *Escherichia coli* strainfor salts **9**, **11**, **12**, **14**, and **15**.

## Data Availability

The data presented in this study are available in this article.

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
