# Peer review of "N-N(+) Bond-Forming Intramolecular Cyclization of O-Tosyloxy β-Aminopropioamidoximes and Ion Exchange Reaction for the Synthesis of 2-Aminospiropyrazolilammonium Chlorides and Hexafluorophosphates"

_ijms, 2023, doi:10.3390/ijms241411315_

Round 1

Reviewer 1 Report

In this study, the authors focused on synthesizing 2-aminospiropyrazolelammonium chlorides and hexafluorophosphates, which are novel compounds with potential applications in drug design and other fields. Through a methodology involving tosylation of β-aminopropioamidoximes with various N-heterocycles, they achieved high yields of 55-97% for the target compounds. The synthesized salts were characterized using physicochemical analysis, IR and NMR spectroscopy, and their structures were confirmed by X-ray diffraction analysis. Although the compounds exhibited weak antimicrobial activity against both gram-negative and gram-positive bacteria in vitro, their unique spirocyclic scaffolds make them valuable for further exploration in diverse applications. The authors need to address some concerns before it is considered for publication:

1. Add more references to the first two paragraphs of introduction.

2. Figure 1: The thermal ellipsoids of two F atoms look larger than the others in salt 14. Is that due to the specific observation perspective or the presence of disorder?

3. Add Rint values to the X-ray data tables in SI. Are the R1 and WR2 for all data or (I > 2σ(I)? Please clarify in the table.

Author Response

Add more references to the first two paragraphs of introduction.

(Q) Add more references to the first two paragraphs of introduction.

(A) In accordance with the requirement of the reviewer 1 it 2 references on the synthesis of azoniaspiro compounds (1) and added 1 reference on the biological activity of azoniaspiro compounds (2) were added.

 (1) P. 2, line 52-57.

Thus, azonia-spiro compounds should be synthesized by reacting the α,ω-dibromoalkane with a cyclic amine (e.g., 1,5-dibromopentane and piperidine to obtain 6-azonia-spiro-[5,5]-undecane [2]. Chiral quaternary N-spiro ammonium bromides with 3',4'-dihydro-1'H-spiro[isoindoline-2,2'-isoquinoline]skeleton were synthesized by reacting of amino alcohols with the corresponding dibromide in acetonitrile or dichloromethane in the presence of DIPEA as a base at r.t. [3].

  1. Millini, R.; Perego, C.; Frigerio, F.; Carluccio, L.; Bellussi, G.L. Azonia-spiro compounds as structure directing agents: a computation study. In: Studies in Surface Science and Catalysis van Steen, L.H. Callanan and M. Claeys (Editors), 2004,154, 275-282. DOI: 1016/S0167-2991(04)80812-7
  2. Bielawski, K.; Leszczyńska,K.; Kałuża,Z.; Bielawska,A.; Michalak,O.; Daniluk,T.; Staszewska-Krajewska,O.; Czajkowska, A.; Pawłowska, N.; Gornowicz, A. Synthesis and antimicrobial activity of chiral quaternary N-spiroammoniumbromides with 3',4'-dihydro-1'H-spiro[isoindoline-2,2'-isoquinoline] skeleton.Drug Des. Devel. Ther. 2017, 11, 2015-2028. DOI: https://doi.org/10.2147/DDDT.S133250

(2) P. 2, line 80-85.

The results of the antibacterial activity study of chiral quaternary N-spiro ammonium bromides with 3',4'-dihydro-1'H-spiro[isoindoline-2,2'-isoquinoline]skeleton showed that gram-negative bacteria were more susceptible towards the tested compounds than gram-positive bacteria; their MICs on gram-negative bacteria typically range in the field 12-100 mg/L; while for gram-positive bacteria it is in the range of 50 ‒>200 mg/L [3].

 (Q) Figure 1: The thermal ellipsoids of two F atoms look larger than the others in salt 14. Is that duet othes pecific observ at ion perspective or the presence of disorder?

(A) This is due to thermal motion of the PF6 anion not involved in strong intermolecular bonding. Although the anion can be indeed disordered over two sites, all refinement parameters increase (with R1 almostequalto 0.1), Alert B concerning low bond precisionon C-C bonds appears, and even isotropic parameters of the disordered fluorine atoms remain very high (see an additional CIF file not for publication). Thus, we decided to deposit to the CSD them odel with high thermal motion of fluorine atoms refined anisotropically.

(Q) Add Rintvalues to the X-ray data table sin SI. Are the R1 and WR2 for all data or (I > 2σ(I)? Please clarify in the table.

(A) Tables S1 and S2 were corrected, these data were corrected.

Reviewer 2 Report

This manuscript presents the preparation and structural determination of some 2-aminospirazolilammonium chlorides and hexafluorophosphates and their biological activity. The spiroammonium part was known, but the hexafluorophosphates and the procedure to obtain them are new.

The chemistry is similar to that described in previous papers of the same research group but contains enough novelty to grant publication.

There are, however, some improvements to be made before the manuscript is publishable.

From the beginning of the text:

Abstract:

Line 22 “2-aminospiropyrazolelammonium…” should be “2-aminospiropyrazolilammonium…”

Line 31: Same issue.

The introduction is confusing. Pyrazolines are important, but the compounds studied here are pyrazolinium cations and have little in common with diphenylpyrazoline. Also, this compound is written as one word or two in different lines (43 and 45).

In Scheme 1, it says “ammoium” instead of ammonium in the structure and the caption.

Line 62: It says that this manuscript is a review, not an article.

The compounds mentioned in section 1.1 (fenspiride, fluspirilene ) are spiro compounds with carbon at the spiro center. Trospium do have a nitrogen at the spiro center (by the way, it is written as Trospiumis)

Compounds a and b of Scheme 2 are compounds 10 and 5 of the manuscript.

These two examples are the only ones pertaining to the background of the manuscript.

Sections 1.2 to 1.5 are interesting but with little in common with the manuscript.

Line 128, there is an extra dash between compound and 5-azonia…

Line 191: “p-touolsulfochloride” should be “p-toluolsulfochloride”

Reference [36] seems to be wrong. It is also the same as the reference [42]

Reference [38] is related to computational calculations, not to X-Ray data.

Line 216 says that compounds 10 and 11 were obtained earlier, but line 228 says that 9, 10, and 12-16 are new.

9, 10, and 11 are known in the literature according to a database search. 9 shows up as being described in Bull Soc Chim France 1976, (3-4), 476.

Line 237, “fix” should be “notice” since the authors did not fix the diastereotopic protons. It is the structure of the molecule as indicated below.

Line 270: reference [41] should be checked for relevance.

Line 279: “nystatin is active…” but according to Table 2, is Gentamicin the active compound.

Line 300 “realizes” does not seem to be the right word. Maybe “…ring is found in the envelope conformation…”

Line 311 The first author of that paper is Delori, not Depori.

In Materials and Methods, the solvent used in NMR should be indicated.

Line 353: “to pH=2” Do this means that ethereal HCl was added until pH=2 is reached? Same question in line 359. It should be written more clearly.

Line 356: the yields should be 68 – 70% since as written there are two yields for three compounds.

In the references section, most of the references have the title of the paper, but others do not have it. There should be consistency in the presentation of the references. 

Author Response

Abstract:

  • Line 22 “2-aminospiropyrazolelammonium…” should be “2-aminospiropyrazolilammonium…”
  • corrected

(Q)Line 31: Same issue.

(A) corrected

(Q) The introduction is confusing. Pyrazolines are important, but the compounds studied here are pyrazolinium cations and have little in common with diphenylpyrazoline. Also, this compound is written as one word or two in different lines (43 and 45).

(A) Pyrazolines are important pharmacophore units and have found practical applications. But due to the fact that there is no information about practically useful properties for pyrazolinium compounds, we, as medical chemists who operate with the concept of “pharmacophore fragment” and, based on the relationship of pyrazoline and pyrazolinium fragments, can assume that there is a possibility of finding useful properties and in the pyrazolinium series.

 (Q) Line 43 и 45: Different spelling: "separate" and "fused" for the word diphenyl pyrazoline and diphenylpyrazoline.

 (A) This is a trivial fragment notation. The name of the material to which we refer has a separate spelling

of the name of the derivatives of this series: [1] Diphenyl pyrazoline market‒Global Outlook and Forecast

2023-2030. 86 p. https://www.24marketreports.com/chemicals-and-materials/global-diphenyl-pyrazoline-

2022-553. (Accessed on 13April2023). Obviously, in this case, one should adhere to the separate spelling “diphenyl pyrazoline”.

Corrected

(Q) In Scheme 1, it says “ammoium” instead of ammonium in the structure and the caption.

(A) corrected

(Q) Line 62: It says that this manuscript is a review, not an article.

(A) corrected as the following: “In this regard, the introduction is mainly based on data on azoniaspirocompounds with parallel use of known data on spiropyrazolinium compounds which may shed light on possible areas of application of the 2-aminospiropyrazolilammoniums we are developing.”

(Q) The compounds mentioned in section 1.1 (fenspiride, fluspirilene ) are spiro compounds with carbon at the spiro center. Trospium do have a nitrogen at the spiro center (by the way, it is written as Trospiumis

(A) Yes, nitrogen-containing spiro-heterocycles are given here as nitrogen-containing spiro-heterocycles. Three of them (Fenspiride, Fluspirilene, Irbesartan) have a сarbon in the head of bridge. Trospium is an example of a spiro-heterocycle with an ammonium nitrogen atom in the head of bridge.

There is a misspelling of this remedy: Trospiumis. The predicate “is” is written in regular text and repeated 2 times. You need to remove it and the correct spelling will remain: Trospium.

Corrected

(Q) Compounds a and b of Scheme 2 are compounds 10 and 5 of the manuscript. These two examples are the only ones pertaining to the background of the manuscript. Sections 1.2 to 1.5 are interesting but with little in common with the manuscript.

(A) Thanks to the reviewer for the general real presentation of cases in our field of study. As stated in the introduction to the article: “In this regard, the introduction is mainly based on data on azoniaspirocompounds with parallel use of known data on spiropyrazolinium compounds which may shed light on possible areas of application of the 2-aminospiropyrazolilammoniums we are developing.”

 (Q) Line 128, there is an extra dash between compound and 5-azonia…

(A) corrected

(Q) Line 191: “p-touolsulfochloride” should be “p-toluolsulfochloride”

(A) corrected

(Q) Reference [36] seems to be wrong. It is also the same as the reference [42].

(A) Dear Reviewer. We allowed unnecessary repetition of reference [36]. There are no references at all in the text [42]. It has been removed from the References list. The total number of links is reduced by one link.

(Q) Reference [38] is related to computational calculations, not to X-Ray data.

(A) You're right. The link provided is redundant. The link that follows is relevant.

 (Q) Line 216 says that compounds 10 and 11 were obtained earlier, but line 228 says that 9, 10, and 12-16 are new.

(A) - The monohydrate of salt 10 has been obtained by some of us before, but the unhydrous solid form is new. We pointed this out at lines 220-221 and 228.

(Q) 9, 10, and 11 are known in the literature according to a database search. 9 shows up as being described in Bull Soc Chim France 1976, (3-4), 476.

(A) As far as we could understand, no spiro-compounds were published by A. Le Berre & C. Porte at [Bull Soc Chim France 1976, (3-4), 476]. Salt 2 from this publication with the same composition as salt 9 in our manuscript has other FT-IR, NMR and Tmelt (142 vs. 288°C), thus, this is another compounds. Synthesis of 10 has indeed been published before, but its anhydrous solid form and biological properties are new.

(Q) Line 237, “fix” should be “notice” since the authors did not fix the diastereotopic protons. It is the structure of the molecule as indicated below.

(A) corrected

(Q) Line 270: reference [41] should be checked for relevance.

(A) Link checked. This should be number [40]: Kayukova, L.A.; Yergaliyeva, E.M.; Vologzhanina, A.V. Redetermination of 2-amino-8-thia-1,5-diazaspiro[4,5]dec-1-en-5-ium chloride. Acta Cryst.,2022, E78, 164-168. DOI: https://doi.org/0.1107/S2056989022000111

(Q) Line 279: “nystatin is active…” but according to Table 2, is gentamicin the active compound.

(A) Yes, there is an error here. Corrected.

(Q) Line 300 “realizes” does not seem to be the right word. Maybe “…ring is found in the envelope conformation…”

(A) corrected

(Q) Line 311 The first author of that paper is Delori, not Depori.

  • corrected

(Q) In Materials and Methods, the solvent used in NMR should be indicated.

(A) corrected

(Q) Line 353: “to pH=2” Do this means that ethereal HCl was added until pH=2 is reached? Same question in line 359. It should be written more clearly.

(A) corrected

(Q) Line 356: the yields should be 68 – 70% since as written there are two yields for three compounds.

(A) corrected

(Q) In the references section, most of the references have the title of the paper, but others do not have

  1. There should be consistency in the presentation of the references. 
  •  

From the authors:

L.A. Kayukova
